# Correlation between Spot Sign and Intracranial Hemorrhage Expansion on Dual-Phase CT Angiography

**DOI:** 10.3390/diagnostics9040215

**Published:** 2019-12-07

**Authors:** Hyesoo Kim, Ja Hong Goo, Hyo Sung Kwak, Seung Bae Hwang, Gyung Ho Chung

**Affiliations:** 1Medical School, Chonbuk National University, Jeonju-si 54896, Korea; for2ms@gmail.com (H.K.); happygoo4@naver.com (J.H.G.); 2Department of Radiology and Research Institute of Clinical Medicine of Chonbuk National University-Biomedical Research Institute of Chonbuk National University Hospital, Geonji-ro Jeonju-si 54907 20, Korea; sbh1010@jbnu.ac.kr (S.B.H.); chunggh@jbnu.ac.kr (G.H.C.)

**Keywords:** intracerebral hemorrhage, CT angiography, brain

## Abstract

Purpose: Expansion of intracranial hemorrhage (ICH) is an important predictor of poor clinical outcome. ICH expansion can be predicted with a spot sign on computed tomographic angiography (CTA). We aimed to evaluate the correlation between spot signs on CTA and ICH expansion on dual-phase CTA. Methods: Patients with spontaneous ICH between January 2017 and April 2019 who underwent an initial CT, dual-phase CTA, and a subsequent CT were retrospectively identified. ICH expansion was defined as volume growth of >33% or >6 mL. We analyzed the presence and change in size of the spot sign in the first phase and second phase CTA. Also, we divided the morphological status of the spot sign, such as a dot-like lesion or linear contrast extravasation, in the first and second phase CTA. Results: A total of 206 patients, including 38 (18.5%) with ICH expansion and 45 (21.8%) with a spot sign, qualified for analysis. Of patients with a spot sign, 26 (57.8%) had ICH expansion on subsequent CT. Increased size of a spot sign in second-phase CTA was more frequent in the ICH expansion group than in the no-expansion group (96.2% vs. 52.6%, *p* < 0.001). First visualization of a spot sign in the second phase was more common in the no-expansion group than in the ICH expansion group (47.4% vs. 3.8%, *p* < 0.001). The morphological patterns of a spot sign between the two groups were not significantly different. Conclusion: Spot signs on dual-phase CTA have different sizes and morphological patterns. Increased size of a spot sign in the second phase of CTA can help identify patients at risk for ICH expansion.

## 1. Introduction

Spontaneous intracerebral hemorrhage (ICH) accounts for 10%–50% of acute strokes, depending on the population, race, and region being studied. [1,2] ICH is the most severe subtype of stroke, with a mortality rate of 35% at seven days after stroke onset to 59% at one year [1]. Outcomes of ICH are significantly worse than of ischemic stroke, with up to 50% mortality at 30 days [2]. Hematoma size is one of the most important predictors of 30-day mortality of ICH [3]. Hematoma expansion occurs in up to 70% of patients who have ICH documented by computed tomographic (CT) scanning performed within 3 h after the onset of symptoms [4]. Hematoma expansion is highly predictive of neurological deterioration and is an independent predictor of mortality and poor functional outcomes. Hematoma expansion has been defined differently among studies: Relative change (e.g., >30% or >33%) or absolute change (e.g., >6 mL or >12.5 mL) in hematoma volume from baseline CT to follow-up CT, or a combination of both [1].

Accurate and reliable clinical and radiographic predictors of ICH growth are needed. Recently, ref. [2] several studies suggested that contrast extravasation on CT angiography (CTA) is a crucial predictor of hematoma expansion and mortality [4]. CTA is a rapid, non-invasive investigation for patients with ICH that is useful for identifying potentially treatable entities, such as aneurysms and other vascular lesions [2]. The spot sign, described as a 1–2 mm focus of contrast enhancement or the presence of high-density material on CTA within an acute primary hematoma, regardless of its shape [1], has been regarded as a predictor of hematoma expansion and risk factor of death or poor outcome [5]. A systematic review and meta-analysis evaluating the accuracy of a spot sign found that CTA identified a spot sign with a sensitivity of 63% and a specificity of 90% [1]. Many CT angiography studies have reported an increase in both spot sign detection and sensitivity in the prediction of hematoma expansion compared with that of single-phase CT angiography [6,7]. Compared with single-phase CTA, multiple-phase CTA provides similar image quality, better vascular enhancement, more hemodynamic information [8] in a faster and easier manner, with lower doses of radiation and contrast material [9].

We hypothesized that combining the spot sign and ICH expansion in dual-phase CTA, as well as the size and shape of the spot sign (such as morphological patterns), could improve the sensitivity for predicting hematoma expansion and poor outcome. This information would be helpful in evaluating clinical prognosis, using the spot sign as a diagnostic criterion.

## 2. Methods

The study protocol was approved by our institutional review board (CNU 2018-10-015-01, 04-11-2018). The requirement for patient informed consent was waived for review of patient records and images.

### 2.1. Patients

This was a retrospective analysis of our institutional data of all patients who were admitted for treatment of ICH from January 2017 to April 2019. During this period, 279 patients were admitted for treatment of ICH. Inclusion criteria for this study were: (1) Age ≥ 18 years; (2) spontaneous ICH in history; (3) immediate non-contrast CT (NCCT) within 1 h after symptom onset; (4) dual-phase CTA examination after NCCT; and (5) follow-up NCCT examination between 2 and 6 h after CTA. Exclusion criteria were: (1) Brainstem or cerebellar hemorrhage; (2) trauma-related hemorrhage; (3) secondary ICH, such as tumor, vasculitis, moyamoya disease, or venous infarction; (4) history of lobar infarction; and (5) previous brain surgery. Also, we performed the follow-up MR imaging in patients with suspicious vascular lesion such as micro-aneurysm or micro-arteriovenous malformation. Of the 279 patients, 206 with spontaneous ICH and complete NCCT and dual-phase CTA examination according to our study protocol were enrolled (Figure 1).

### 2.2. CT Acquisition

Initial NCCT (Definition Flash; Siemens, Erlangen, Germany), with a slice thickness of 1.0 mm, were obtained for all patients. Patients in our sample had a first-phase CTA if ICH was suspected on NCCT. First-phase CTA was performed by scanning from the cerebral vertex to the aortic arch with 0.7 mm slices. The first-phase acquisition was timed to occur during the peak arterial phase. The second phase was performed immediately after completion of the first-phase acquisition and covered from the base of the skull to the cerebral vertex. Nonionic contrast media (80–120 mL) was administered into the antecubital vein at 3–5 mL/s, and the CTA source images for evaluation of atherosclerosis or vascular malformation were post-processed and reformatted to create coronal, sagittal, and axial multiplanar images. Follow-up NCCT with a slice thickness of 1.0 mm was performed, except for patients who needed emergency operation.

### 2.3. Clinical Data

Clinical and demographic data were acquired through retrospective review of medical charts. 

The collected data included sex, age, underlying disease, previous medication, and initial Glasgow Coma Scale for correlation between hematoma expansion and imaging findings of NCCT and CTA.

### 2.4. Image Analysis

CTA images for diagnosis of spot sign were reviewed retrospectively by a neuroradiologist with 25 years of experience. He was not informed about the composition of the patient population, and he independently evaluated the anonymized, randomized images. Spot sign on CTA was defined as presence of at least 1 focus of contrast density within the ICH, without connection to normal or abnormal vessels surrounding the hemorrhage, and without hyper density at the corresponding location on NCCT [10,11]. The reviewer reviewed dual-phase CTAs according to standard radiological viewing. The following characteristics of the spot sign in each phase on CTA were recorded: Total number of spot signs, spot-size change on dual-phase, and linear or dot-like patterns of spot signs. Linear spot sign was defined as high density, more than 10 mm long on CTA. If the spot signs found was more than two, the reviewer chose the longer and larger spot sign for analysis.

Another reviewer measured the volume of ICH on initial and follow-up NCCT in millimeters, using the ABC/2 method [12]. An increase of hematoma size >33% or >6 mL was considered significant enlargement [13,14].

### 2.5. Statistical Analysis

Patients who had a spot sign were divided into 2 groups according to presence of ICH expansion. Continuous variables were expressed as means with standard deviation, while categorical data were expressed as counts and percentages. The independent t-test was used to analyze differences in continuous variables, and Pearson’s chi-squared test was used for categorical variables. The statistical significance level was set at *p* < 0.05. Statistical analyses were performed with SPSS v. 23.0 for Windows (IBM, Somers, NY, USA).

## 3. Results

In total, 206 patients with spontaneous ICH, complete NCCT, and dual-phase CTA were enrolled. The demographic characteristics of patients according to presence of ICH expansion are summarized in Table 1. Thirty-eight (18.4%) of the patients had ICH expansion that was seen on immediate follow-up NCCT. Forty-six patients (22.3%) (median age, 65.5 years; age range, 32–88 years; male, 62.2%) had a positive spot sign. Of these 46 patients, 26 (57.8%) had ICH expansion seen on serial follow-up CT scan. Of 46 patients with spot sign, initial ICH volume in patients with ICH expansion was higher than that in patients without ICH expansion, although the difference was not statistically significant (27.6 ± 23.9 vs. 20.8 ± 15.3, *p* = 0.274).

Interpretation of dual-phase CTA and correlation of ICH expansion in patients with abnormal CT findings are summarized in Table 2. Of 46 patients with spot sign, the sign was present in 35 (76.1%) on the first phase on CT, and the sign increased in size on the second phase (Figure 2). Of 26 patients with ICH expansion, the sign was seen in 25 (96.2%) in the first phase with increased size in the second phase. First-phase visualization of the spot sign and increase size of the sign in the second phase was significantly higher in patients with ICH expansion than in those without ICH expansion (96.2% vs. 50.0%, *p* < 0.001) (Figure 3). Initial visualization of the spot sign on the second phase on CTA was significantly higher in patients without ICH expansion (*p* < 0.001) (Figure 4). However, all patients with visualization of spot sign on first phase on CTA showed size increase on second phase regardless of ICH expansion (25/25 patients with ICH expansion and 10/10 patients with no ICH expansion). The morphological patterns of the spot sign that linear or dot-like were not significant different. Two patients without ICH expansion had two dot-like spot signs on first-phase CTA.

## 4. Discussion

The main finding of this retrospective study is that spot signs on dual-phase CTA had different sizes and morphological patterns. Specifically, spot signs that were visualized in the first phase increased in the second phase on CTA for most patients. Also, our study divided the morphological status of the spot sign, such as a dot-like lesion or linear contrast extravasation, in the first and second phase on CTA. The main result of our study demonstrated that if spot signs visualized in the first phase and spot size increased in the second phase on CTA, that can help stratify patients at risk for ICH expansion. However, the results suggest that the morphological pattern of spot sign does not have clinical significance.

Clinically significant hematoma expansion, occurring in up to one-third of patients with ICH, has been identified as one of the most important determinants of early neurological deterioration and poor clinical outcomes in spontaneous ICH patients [15]. Hematoma growth is thought to be due to active hemorrhage and rebleeding and is reportedly an independent determinant of mortality and morbidity [16]. Hematoma growth has been reported in 38% of patients after initial CT [17]. Because of the rapid and severe devastation associated with ICH, predictors of potential hematoma expansion could help clinicians stratify patients timely and effectively. In our study, 18.4% (38/206) of the patients had ICH expansion that was seen on immediate follow-up NCCT.

Prediction scores have recently been developed and validated for prediction of hematoma expansion [18]. Sheng et al. [18] reported individual predictors of hematoma expansion, which include clinical (GCS, National Institutes of Health stroke scale (NIHSS), blood pressure, and blood glucose), laboratory (anticoagulation, inflammation, and microvascular integrity), and radiographic (interval time from onset, the baseline ICH volume, the shape and density on CT, intraventricular hemorrhage, spot sign parameters, and modified spot signs) variables, as well as prediction scores (9-point, BRAIN, PREDICT A/B) [18]. In our study, there were differences concerning gender (m 55.3% vs. 67.2%), chronic renal disease (7.9% vs. 5.4%), liver disease (15.8% vs. 10.7%), alcoholics (39.5% vs. 30.4%), and antiplatelet or anticoagulation drugs, but that they were not significant.

Early CT protocols in patients with ICH used only NCCT or CTA after NCCT for evaluation of critical imaging findings of ICH expansion [19]. Subsequent papers reported the relationship between imaging findings of CTA and NCCT and ICH expansion [20,21,22]. The study of Sporns et al. [20] found that NCCT imaging markers were strongly correlated with an established spot sign. Taking into account that a spot sign has a higher sensitivity for outcome prediction than does a black hole sign or blend sign alone, both NCCT and CTA should be acquired if possible [20]. The results of other studies [21] have suggested that CT variables indicating early hematoma expansion, such as spot sign on CTA, are the most reliable predictors of outcome. Also, a spot sign had a higher sensitivity for predicting outcome than did black hole sign in any location (34.6% vs. 23.1%) [22].

David et al. [23] reported the predictive value of ICH with multiphase CT angiography. They demonstrated that the later the phase of multiphase CT angiography, the higher the frequency of the spot sign in patients with acute ICH (29.3% in phase 1, 43.1% in phase 2, 46.3% in phase 3; *p* < 0.001). However, our study was based on the dual-phase CT angiography protocol, so the second phase was performed immediately after the first phase acquisition. Shin et al. [24] empirically found that dual-phase CT is helpful for predicting clinical outcome in patients with acute stroke. We hypothesized that rapid dual-phase CTA is more efficient than multiphase CTA, considering the shorter time of imaging acquisition. Multiphase CTA has advantages over single-phase CTA, but the long time it requires is a drawback in emergent situations. Therefore, in our study, dual phase CT for evaluation of spot signs showed the different size and patterns between the first and second phases.

Andrew et al. [25] identified four different patterns of spot sign appearance and speculated about potential vascular pathology. According to those authors, spot signs have a serpiginous and/or spot-like appearance, may be multiple or single, are greater than 1.5 mm, occur within the hematoma margin without connection to outside vessels, and have a density twice that of the background hematoma density. Whether the different spot sign patterns or the number of spot signs within a hematoma will provide additional prognostic value for prediction of the extent of hematoma expansion remains to be evaluated from the final PREDICT/Sunnybrook study patient cohort [25]. Whether the different patterns indicate different lesion grades with varying risk of further extravasation or hematoma enlargement will be determined in a larger prospective CTA ICH dataset and future spot sign-based clinical trials. In our study, we analyzed the spot sign pattern by a linear or dot like pattern between ICH expansion and no-ICH expansion groups due to small patient number. There was no clinical significance between the morphological patterns of the spot sign.

Some limitations should be considered in the interpretation of our results. First, although based on a well-characterized ICH group, our findings derive from a nonrandomized, single-center retrospective analysis. Therefore, multicenter studies are needed to further evaluate predictive value. Second, a proportion of patients were excluded from the analysis because of missing CTA images, indicating a potential bias. Follow-up CTA may not have been performed because of withdrawal of care or death within the first 24 h. Third, CT angiography is contraindicated in patients with prior contrast reactions and renal impairment. Further studies need to compare sensitivity and specificity between the spot sign and all NCCT signs. Fourth, we did not perform the digital subtraction angiography (DSA) for exclusion of spot sign mimickers such as microvascular aneurysm, micro-arteriovenous malformation, or fibrin globules. DSA can be used to aid in identification of the vascular source and avoid misdiagnosing spot sign mimickers for an ICH spot sign. In our study, we performed the follow-up MR imaging study in patients with suspicious vascular lesion. Finally, we analyzed the imaging findings related to ICH expansion, but did not including mortality rates or clinical outcome.

## 5. Conclusions

Dual-phase CT angiography is a useful tool for predicting hematoma expansion. Spot signs on dual-phase CTA have different sizes and morphological patterns, but they seem to be not relevant for predicting ICH expansion. Although spot sign is relatively rare in patients with spontaneous ICH, its presence is predictive for ICH expansion.

## Figures and Tables

**Figure 1 diagnostics-09-00215-f001:**
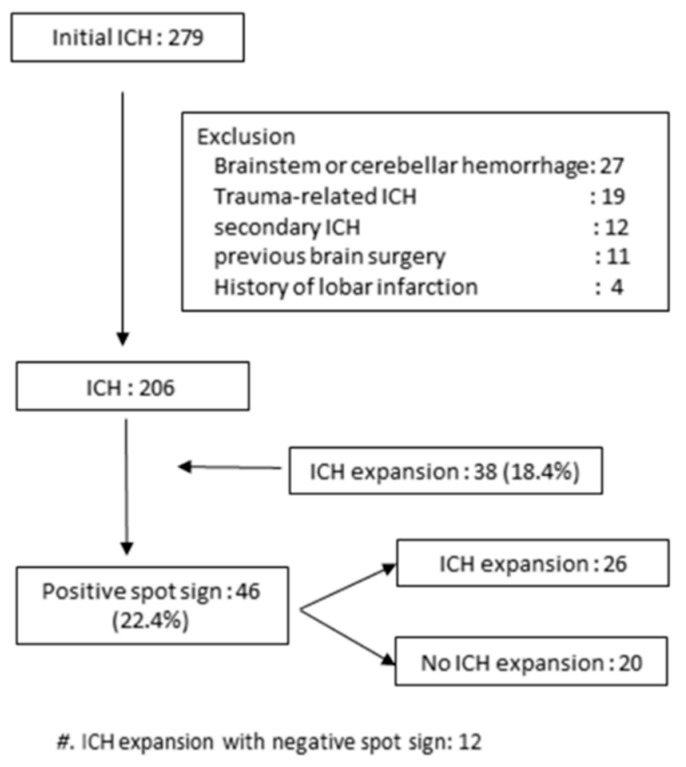
Flow diagram of patient selection.

**Figure 2 diagnostics-09-00215-f002:**
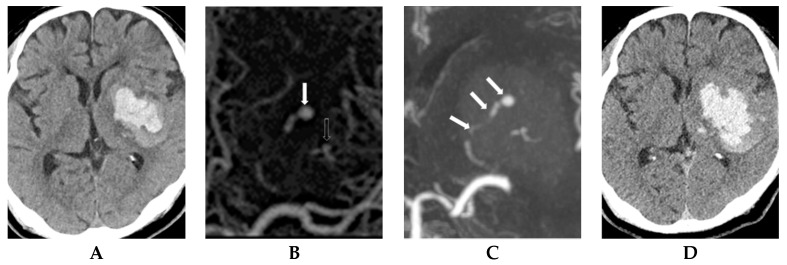
A 70-year-old man with left hemiplegia. (**A**) Initial non-contrast CT scan shows an intracranial hemorrhage in the left basal ganglia. (**B**) The first phase on CT angiography shows a positive spot sign with a linear (close arrow) and dot-like pattern (open arrow) in the hemorrhage. (**C**) The second phase on CT angiography shows an increased size of linear-like spot sign (arrows). (**D**) Follow-up CT scan after 3 h shows intracranial hemorrhage expansion. Follow-up MR examination did not show any aneurysmal lesion in hemorrhagic parenchyma.

**Figure 3 diagnostics-09-00215-f003:**
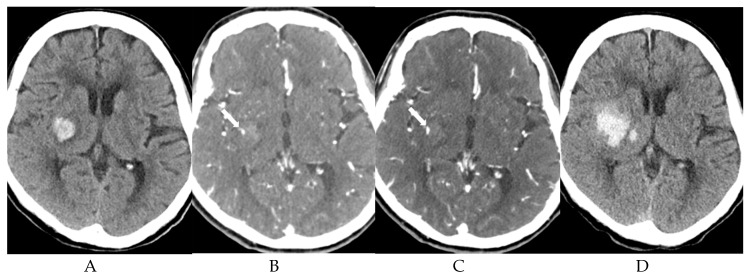
An 84-year-old woman with left hemiplegia. (**A**) Initial non-contrast CT scan shows an intracranial hemorrhage in the right basal ganglia. (**B**) The first phase on CT angiography shows a positive spot sign with a small dot-like pattern in the hemorrhage (arrow). (**C**) The second phase on CT angiography shows an increased size of spot sign (arrow). (**D**) Follow-up CT scan after 2 h shows intracranial hemorrhage expansion.

**Figure 4 diagnostics-09-00215-f004:**
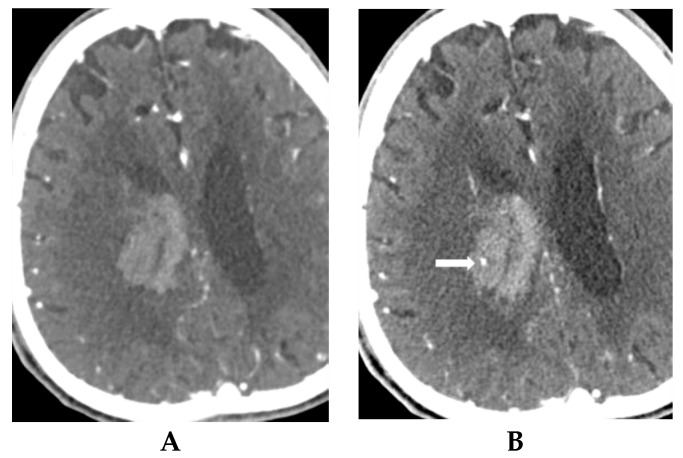
An 80-year-old man with left hemiplegia and right basal ganglia hemorrhage. (**A**) The first phase on CT angiography does not show any spot sign. (**B**) The second phase on CT angiography shows a positive spot sign with a small dot-like pattern at the upper margin of hemorrhage (arrow). Follow-up CT scan after 3 h does not show any change in intracranial hemorrhage volume.

**Table 1 diagnostics-09-00215-t001:** Demographic characteristics of patients according to presence of intracranial hemorrhage (ICH) expansion.

	ICH Expansion (*n* = 38)	No ICH Expansion (*n* = 168)	*p*
Age	64.7 ± 14.3	66.5 ± 12.9	0.448
Male (%)	21 (55.3)	112 (67.2)	0.166
Hypertension (%)	18 (47.4)	79 (47.0)	0.965
Diabetes (%)	11 (28.9)	58 (34.5)	0.510
Chronic renal disease (%)	3 (7.9)	9 (5.4)	0.555
Liver disease (%)	6 (15.8)	18 (10.7)	0.377
Cardiac disease (%)	5 (13.5)	31 (18.5)	0.466
Current smoking, (%)	6 (15.8)	39 (23.2)	0.320
Alcoholic, (%)	15 (39.5)	51 (30.4)	0.279
Antiplatelet drug (%)	5 (13.5)	17 (10.1)	0.542
Anticoagulant drug (%)	3 (7.9)	8 (4.8)	0.445
ICH volume	27.6 ± 23.9	23.2 ± 14.8	0.147
IVH presence, (%)	16 (42.1)	81 (48.2)	0.497

ICH—intracerebral hemorrhage; IVH—intraventricular hemorrhage.

**Table 2 diagnostics-09-00215-t002:** Interpretation of dual-phase computed tomographic angiography (CTA) and correlation of ICH expansion in patients with abnormal computed tomographic (CT) findings.

Spot Sign	ICH Expansion (*n* = 26)	No ICH Expansion (*n* = 20)	*p*
First phase visualization	25 (96.2)	10 (50.0)	<0.001
Increased size on second phase	25 (96.2)	10 (50.0)	<0.001
First visualization on second phase	1 (3.8)	10 (50.0)	<0.001
Linear pattern	14 (53.8)	9 (45.0)	0.675
Dot-like pattern	12 (52.6)	11 (55.0)	0.675

ICH—intracerebral hemorrhage; CTA, CT angiography.

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
