# Peer review of "Correlation between Spot Sign and Intracranial Hemorrhage Expansion on Dual-Phase CT Angiography"

_diagnostics, 2019, doi:10.3390/diagnostics9040215_

Round 1
Reviewer 1 Report
Hyesoo Kim and colleagues present an interesting investigation about 'spot sign' on dual-phase CT and its predictive value for ICH progression.
Although, the methods are elaborate and well done there are some methodical and structural errors in this research article.
1) First of all: if there are references indicating any incidences or prevalences (e.g. ref [1], line 35-37), the given reference should refer to the origin article and not to a 'reference of a reference'. So please have a closer look throughout the article, where numbers come from.
2) How did the authors ensure that aneurysm bleedings or micro-AVM are excluded? Because aneurysms for example can imitate spot signs. Peng et al. reported some ‘spot sign mimickers’ in their article [1].
3) In the result chapter it is hard to retrace how the study population was divided: 206 total, 38 with ICH expansion, 46 with spot sign - and the rest? Perhaps a fourfold table is helpful.
4) The percentage value in line 116 isn't right for my opinion (46/206 = 22.3 %).
5) Table 1 has to be renamed, because it shows characteristics of the 46 patients with spot sign and not the characteristics of the whole study population – although this would be indeed interesting in addition.
6) Line 132: "The morphological patterns of the spot sign that linear or dot-like were similar in the 2 groups" better redraft as "were not significant different", because they are different.
7) Figure 1: Please provide a measurement, because vessels also look different in size between picture B and C. "Dot like" and "linear pattern" should be indicated with different arrows. Furthermore, for my opinion an aneurysm bleeding could not be excluded based on these pictures.
8) Figure 2: For my opinion the pictures or description are sorted wrong: A – C - B – D is maybe correct. Unfortunately, the pictures A/B and C/D show different slicing planes and are therefore only comparable with restriction.
9) Figure 3: For my opinion the pictures are also sorted wrong – or the description is wrong. And again different slicing planes.
10) The discussion part isn't really a discussion of the results. Thus, two thirds of the text give a summary of the literature (lines 170-185, 187-203, 212-228) and the authors don't argue in context with presented literature results. For example:
-line 162: The authors state “The main finding of this retrospective study is that spot signs … had different sizes and morphological pattern”. This wasn’t the aim of the investigation and is controversial to the statement in line 132 “The morphological patterns of the spot sign … were similar in the 2 groups”
-line 170 following they report that ICH expansion is associated with poor clinical outcome. How was it in the study population? Is there a correlation with clinical outcome parameters (e.g. mortality at 3 months)? Was the clinical outcome measured? Why not?
-line 174: "Hematoma growth has been reported in 38% of patients after initial CT.[17]" - please correlate with the study population and discuss.
-line 179 following they also report criteria associated with poor clinical outcome (blood pressure, blood glucose, laboratory, radiographic interval...). How was it in the study population? For me it is interesting that 'liver disease' and 'alcoholic' were higher in the expansion group. Could any anticoagulation disorder be excluded? Because that might be a cause for ICH progression too.
-line 179 following: ICH volume was also reported to be a predictive item (line 183) and in the study population the ICH volume was higher in the expansion group! Please discuss. Is this perhaps predictive? Perhaps ICH volume correlate with ICH progression even in the other patients without spot sign?
-line 185: “The clinical and individual characteristics of our study population are summarized in Table 1.” This have to be corrected, thus table 1 shows only characteristics of the 42 patients with ICH and spot sign – and not the characteristics of the 206 patients.
-line 187: “In our study, clinical and laboratory predictors are independent of previously described predictors of outcome for symptomatic ICH. [19-22]” This statement is not provable with the current results. They don’t present any laboratory or clinical results. And why is there a reference denoted, if this are results of the own study?
-Line 198: “Many studies have documented that the CTA spot sign is a good predictor … for small hematoma volumes [26]” – please correlate with the study population and discuss. For me that is the clue of spot sign: It is relevant for small bleedings that must be closely monitored for expansion because it might have impact on clinical outcome. With increase of ICH volume spot sign forfeit its predictive property because prognosis is devastating anyway.
-Line 220: Andrew et al reported four different patterns of spot signs. Why was this classification not used in this study? Number and patterns of spot signs were documented. So, discuss, because “…the number of spot signs within a hematoma, will provide additional prognostic value for prediction of the extent of hematoma expansion…”.
11) Line 198: “BHS” is not explained.
12) Line 202 following the authors suddenly introduce a new aim of their study: line 207 “We hypothesized that rapid dual-phase CTA is more efficient than multiphase CTA” and line 209 “Our study suggests that using dual-phase CTA may be sufficient in predict hematoma expansion of most cases” These aims weren’t established before in the introduction. The authors should clearly state their aims in the introduction.
13) In the conclusion the authors state that “The spot sign on CT angiography predicts hematoma expansion and may be related to mortality and functional outcomes”. This for my opinion could not be the conclusion, because the authors neither present any mortality data nor any functional outcome parameters. Brouwers et al. (JAMA Neurol “Predicting hematoma expansion after primary intracerebral hemorrhage”) for example present a scoring system that should be included.
14) Conclusions for me are:
-spot sign is rare in ICH, about one fifth, prevalence 22.3 % (46/206)
-prevalence of spot sign is predictive for ICH expansion – YES – significant
-spot sign first visualized in second phase is predictive for NO ICH expansion (table 2) – significant
-increase of spot sign is NOT predictive for ICH expansion --> Statement: spot sign increased in 25/25 patients with ICH expansion and in 10/10 patients with no ICH expansion. In both groups 100 % (!) showed increase of spot sign. The difference must be significant because it is the same population/ number like in “first phase visualization” that is tested for significance. It is possible but illogical to give a p-value.
Also Peng et al. [1] said: “I think despite spot sign success in predicting HE, spot sign has not been viewed as a gold standard to predict HE in ICH patients.”
15) Using a fourfold table (s. attachment) sensitivity (0.4), specificity (0.86) and a positive predictive value (ppv, 0.56) of spot sign can calculated. According these results the predictive value of spot sign is like tossing a coin.
|
|
Spot sign positive |
Spot sign negative |
|
|
ICH expansion |
26 |
38 |
64 |
|
No ICH expansion |
20 |
122 (?) |
142 |
|
|
46 |
160 |
206 |
Author Response
First of all: if there are references indicating any incidences or prevalences (e.g. ref [1], line 35-37), the given reference should refer to the origin article and not to a 'reference of a reference'. So please have a closer look throughout the article, where numbers come from. Answer) We added references in this sentence. How did the authors ensure that aneurysm bleedings or micro-AVM are excluded? Because aneurysms for example can imitate spot signs. Peng et al. reported some ‘spot sign mimickers’ in their article [1].
Answer) We added this sentence about your comment.
Patients section : Also, we was performed the follow-up MR imaging in patients with suspicious vascular lesion such as micro-aneurysm or micro-arteriovenous malformation.
Discussion limitation paragraph.: Fourth, we did not perform the digital subtraction angiography (DSA) for exclusion of spot sign mimickers such as microvascular aneurysm, micro-arteriovenous malformation, or fibrin globules. DSA can be used to aid in identification of the vascular source and avoid misdiagnosing a spot sign mimickers for an ICH spot sign. In our study, we performed the fallow up MR imaging study in patients with suspicious vascular lesion
In the result chapter it is hard to retrace how the study population was divided: 206 total, 38 with ICH expansion, 46 with spot sign - and the rest? Perhaps a fourfold table is helpful. Answer) Our paper was focused the correlation with spot sign on CTA angiography and ICH expansion. So, first we selected the patients with spot sign (N = 46). Therefore, remained patients did not had the spot sign. We excluded this patients for out study. The percentage value in line 116 isn't right for my opinion (46/206 = 22.3 %). Answer) Sorry, We changed. Table 1 has to be renamed, because it shows characteristics of the 46 patients with spot sign and not the characteristics of the whole study population – although this would be indeed interesting in addition. Answer) Yes, we changed this table 1.
Table 1. Demographic characteristics of patients according to presence of ICH expansion.
line 132: "The morphological patterns of the spot sign that linear or dot-like were similar in the 2 groups" better redraft as "were not significant different", because they are different. Answer) We changed this sentence. Figure 1: Please provide a measurement, because vessels also look different in size between picture B and C. "Dot like" and "linear pattern" should be indicated with different arrows. Furthermore, for my opinion an aneurysm bleeding could not be excluded based on these pictures. Answer) We inserted the different arrows in Figure 1B. This patient did not show any aneurysmal component on follow up MR examination. So, we excluded the aneurysmal rupture. Therefore, we described in the Figure 1. Figure 2: For my opinion the pictures or description are sorted wrong: A – C - B – D is maybe correct. Unfortunately, the pictures A/B and C/D show different slicing planes and are therefore only comparable with restriction.
Answer) We changed the figure numbering. Also, we should use the different imaging slice due to ICH expansion and increased size of spot sign.
Figure 3: For my opinion the pictures are also sorted wrong – or the description is wrong. And again different slicing planes.
Answer) We changed the figure numbering.
The discussion part isn't really a discussion of the results. Thus, two thirds of the text give a summary of the literature (lines 170-185, 187-203, 212-228) and the authors don't argue in context with presented literature results. Answer) We changed this paragraph and deleted the last paragraph (line 212 – 228) For example:-line 162: The authors state “The main finding of this retrospective study is that spot signs … had different sizes and morphological pattern”. This wasn’t the aim of the investigation and is controversial to the statement in line 132 “The morphological patterns of the spot sign … were similar in the 2 groups”
Answer) This sentence is mean that spot sign had different status on dual phase CT. We described our result in the final sentence of this paragraph.
12)-line 174: "Hematoma growth has been reported in 38% of patients after initial CT.[17]" - please correlate with the study population and discuss.
Answer) We described our result in this paragraph.
In our study, 18.4% (38/206) of the patients had ICH expansion that seen on immediate follow-up NCCT.
13) -line 179 following they also report criteria associated with poor clinical outcome (blood pressure, blood glucose, laboratory, radiographic interval...). How was it in the study population? For me it is interesting that 'liver disease' and 'alcoholic' were higher in the expansion group. Could any anticoagulation disorder be excluded? Because that might be a cause for ICH progression too. line 179 following: ICH volume was also reported to be a predictive item (line 183) and in the study population the ICH volume was higher in the expansion group! Please discuss. Is this perhaps predictive? Perhaps ICH volume correlate with ICH progression even in the other patients without spot sign?
Answer) We discuss the risk of hematoma expansion and the prognosis factors in this paragraph. It was written to show the basis for the research of dual phase CT, the main purpose of our study. Also, we described our result in last sentence.
14) -line 185: “The clinical and individual characteristics of our study population are summarized in Table 1.” This have to be corrected, thus table 1 shows only characteristics of the 42 patients with ICH and spot sign – and not the characteristics of the 206 patients.
Answer) We changed Table 1 about your comment.
15) -line 187: “In our study, clinical and laboratory predictors are independent of previously described predictors of outcome for symptomatic ICH. [19-22]” This statement is not provable with the current results. They don’t present any laboratory or clinical results. And why is there a reference denoted, if this are results of the own study?
Answer) We deleted references.
16) -Line 198: “Many studies have documented that the CTA spot sign is a good predictor … for small hematoma volumes [26]” – please correlate with the study population and discuss. For me that is the clue of spot sign: It is relevant for small bleedings that must be closely monitored for expansion because it might have impact on clinical outcome. With increase of ICH volume spot sign forfeit its predictive property because prognosis is devastating anyway.
Answer) We deleted this sentence due to some confuse of our study.
17) -Line 220: Andrew et al reported four different patterns of spot signs. Why was this classification not used in this study? Number and patterns of spot signs were documented. So, discuss, because “…the number of spot signs within a hematoma, will provide additional prognostic value for prediction of the extent of hematoma expansion…”.
Answer) We divided simple pattern due to small patient’s number. So, we described in the paragraph.
18) Line 198: “BHS” is not explained.
Answer) We deleted this sentence.
19) Line 202 following the authors suddenly introduce a new aim of their study: line 207 “We hypothesized that rapid dual-phase CTA is more efficient than multiphase CTA” and line 209 “Our study suggests that using dual-phase CTA may be sufficient in predict hematoma expansion of most cases” These aims weren’t established before in the introduction. The authors should clearly state their aims in the introduction.
Answer) We changed this sentence.
“David et al [23] reported the predictive value of ICH with multiphase CT angiography. They demonstrated that the later the phase of multiphase CT angiography, the higher the frequency of the spot sign in patients with acute ICH (29.3% in phase 1, 43.1% in phase 2, 46.3% in phase 3; p<0.001). However, our study was based on the dual-phase CT angiography protocol, so the second phase was performed immediately after the first phase acquisition. Shin et al [24] empirically found that dual-phase CT is helpful for predicting clinical outcome in patients with acute stroke. We hypothesized that rapid dual-phase CTA is more efficient than multiphase CTA, considering the shorter time of imaging acquisition. Multiphase CTA has advantages over single-phase CTA, but the long time it requires is a drawback in emergent situations. Therefore, in our study, dual phase CT for evaluation of spot sign showed the different size and patterns between first and second phase. “
22) In the conclusion the authors state that “The spot sign on CT angiography predicts hematoma expansion and may be related to mortality and functional outcomes”. This for my opinion could not be the conclusion, because the authors neither present any mortality data nor any functional outcome parameters. Brouwers et al. (JAMA Neurol “Predicting hematoma expansion after primary intracerebral hemorrhage”) for example present a scoring system that should be included.
Answer) We deleted this sentence.
23) Conclusions for me are:
-spot sign is rare in ICH, about one fifth, prevalence 22.3 % (46/206)
-prevalence of spot sign is predictive for ICH expansion – YES – significant
-spot sign first visualized in second phase is predictive for NO ICH expansion (table 2) – significant -increase of spot sign is NOT predictive for ICH expansion --> Statement: spot sign increased in 25/25 patients with ICH expansion and in 10/10 patients with no ICH expansion. In both groups 100 % (!) showed increase of spot sign. The difference must be significant because it is the same population/ number like in “first phase visualization” that is tested for significance. It is possible but illogical to give a p-value.
Also Peng et al. [1] said: “I think despite spot sign success in predicting HE, spot sign has not been viewed as a gold standard to predict HE in ICH patients.”
Answer) we changed the conclusion section about your comment.
24) Using a fourfold table (s. attachment) sensitivity (0.4), specificity (0.86) and a positive predictive value (ppv, 0.56) of spot sign can calculated. According these results the predictive value of spot sign is like tossing a coin.
|
|
Spot sign positive |
Spot sign negative |
|
|
ICH expansion |
26 |
38 |
64 |
|
No ICH expansion |
20 |
122 (?) |
142 |
|
|
46 |
160 |
206 |
|
|
|
|
|
Answer) Purpose of our study was analyzed the imaging findings of dual phase CT in patients with ICH. Therefore, we focused the imaging findings of dual phase CT. We sincerely ask for your understanding of this point. Insertion of this table is inconsistent with our purpose.
Reviewer 2 Report
The authors investigated the efficiency of dual-phase CT angiography in ICH. This manuscript is well written but has some issues.
The biggest issue is that similar investigations for dual-phase or three-phase CTA in hemorrhagic stroke have previously been reported; for example, Delgado Almandoz et al. (J Neurointerv Surg 2012), Tsukabe et al. (Neuroradiology 2014), and Wu et al. (Acta Radiol 2018) for ICH and Suzuki et al. (World Neurosurgery 2019) for SAH. The authors should cite these previous articles and discuss the novelty of their study. A morphological investigation may increase the value of the authors’ study.
A minor issue that should be corrected is that the legends for Figures 2B–2D do not match the figures.
Author Response
2-1) A minor issue that should be corrected is that the legends for Figures 2B–2D do not match the figures.
Answer) We changed the figure numbering.
Reviewer 3 Report
In this paper the authors investigated the role of spot-sign as predictor of ICH expansion. Although data on this topic are well known, the authors decided to collect and evaluate new parameters of the spot-sign, such as size and shape, using a dual-phase CTA study. I agree that this decision makes this study novel and interesting. However, the present form of the study is affect by too many concerns that should be carefully considered.
Major concerns:
1) Please, modify the running title underlying the novel aspects of the study, i.e. use of dual-phase CTA and use of spot-sign's size and shape.
2) 73 patients were not included. Please, provide descriptions of reasons for exclusion. Moreover, add a figure with the diagram of the study population.
3) The authors collected information on initial GCS. Since the study is focused on ICH expansion, information on GCS at discharge or short-term mortality is needed. In addition, I ask the authors to provide on NIHSS at admission and at discharge.
4) As correctly reported in the Introduction section, ICH expansion may be defined as an increasing in hematoma size more than 30 or 33% or an absolute change more than 6 or 12.5 ml. The authors should explain the reason of their choice regarding the definition of ICH expansion.
5) Statistical analysis should be re-performed. In particular, the authors should compare the 168 patients with no ICH expansion with the 38 patients with ICH expansion. This new comparison will include demographic findings, clinical data and radiological data. In particular, information on presence, number, size and shape of the spot-sign should be added to this comparison.
Author Response
Reviewer 3)
3-1)Please, modify the running title underlying the novel aspects of the study, i.e. use of dual-phase CTA and use of spot-sign's size and shape.
Answer) We changed the running title about your comment.
3-2)73 patients were not included. Please, provide descriptions of reasons for exclusion. Moreover, add a figure with the diagram of the study population.
Answer) We added the diagram of the study population: Figure 1.
3-3) The authors collected information on initial GCS. Since the study is focused on ICH expansion, information on GCS at discharge or short-term mortality is needed. In addition, I ask the authors to provide on NIHSS at admission and at discharge.
Answer) We did not analyzed the NIHSS score due to focus of correlation between imaging finding of dual phase CT and ICH expansion.
3-4) As correctly reported in the Introduction section, ICH expansion may be defined as an increasing in hematoma size more than 30 or 33% or an absolute change more than 6 or 12.5 ml. The authors should explain the reason of their choice regarding the definition of ICH expansion.
Answer) Recently ICH papers were used as followed: increased of hematoma size >33% or >6 mL. Therefore, we added the references in this sentence.
3- 5) Statistical analysis should be re-performed. In particular, the authors should compare the 168 patients with no ICH expansion with the 38 patients with ICH expansion. This new comparison will include demographic findings, clinical data and radiological data. In particular, information on presence, number, size and shape of the spot-sign should be added to this comparison.
Answer) This comment is similar to comment of reviewer 1. So, we changed this table.
Round 2
Reviewer 1 Report
The manuscript provided by Hyesoo Kim and colleagues has now markedly improved. Introduction and presentation of results are conclusive now. The labeling at the figures are now helpful indeed.
Moderate English language and typing editing is necessary throughout the text (e.g. line 112, 123, 137, 138, 170, 183).
I am sorry, but there are some additional questions:
-Where does the percentage value (52.6 %) in line 135 come from? - for me it is 50 % according to table 2.
-Line 199: BHS is black hole sign?
-Line 190: How could you state that “clinical and laboratory predictors are independent…for ICH expansion”? The categories in table 1 sound like taken from patient’s history. There aren’t any laboratory results presented. Perhaps better state that there were differences concerning gender (m 55.3 vs 67.2 %), chronic renal disease (7.9 vs 5.4 %), liver disease (15.8 vs 10.7 %), alcoholic (39.5 vs 30.4 %), antiplatelet or anticoagulation drugs but that they weren’t significant.
For conclusion I suggest:
"Dual-phase CT angiography is a useful tool for predicting hematoma expansion. Spot signs on dual-phase CTA have different sizes and morphological patterns, but they seem to be not relevant for predicting ICH expansion. Although, spot sign is relatively rare in patients with spontaneous ICH, its presence is predictive for ICH expansion."
"Spot sign first visualized in second phase was predictive for no ICH Expansion in our study." - or are there any study results that support this?
Author Response
Reviewer 1
Rev#1-1) Moderate English language and typing editing is necessary throughout the text (e.g. line 112, 123, 137, 138, 170, 183).
Answer) We changed English Language about your comment.
Rev#1-2) Where does the percentage value (52.6 %) in line 135 come from? - for me it is 50 % according to table 2.
Answer) We changed this percentage.
Rev#1-3) -Line 199: BHS is black hole sign?
Answer) We changed this word.
Rev#1-4) -Line 190: How could you state that “clinical and laboratory predictors are independent…for ICH expansion”? The categories in table 1 sound like taken from patient’s history. There aren’t any laboratory results presented. Perhaps better state that there were differences concerning gender (m 55.3 vs 67.2 %), chronic renal disease (7.9 vs 5.4 %), liver disease (15.8 vs 10.7 %), alcoholic (39.5 vs 30.4 %), antiplatelet or anticoagulation drugs but that they weren’t significant.
Answer) Yes, we changed this sentence as your comment.
Rev#1-5) For conclusion I suggest:
"Dual-phase CT angiography is a useful tool for predicting hematoma expansion. Spot signs on dual-phase CTA have different sizes and morphological patterns, but they seem to be not relevant for predicting ICH expansion. Although, spot sign is relatively rare in patients with spontaneous ICH, its presence is predictive for ICH expansion."
Answer) We changed the conclusion section as your comment.
Rev#1-6) "Spot sign first visualized in second phase was predictive for no ICH Expansion in our study." - or are there any study results that support this?
Answer) We deleted the last sentence.
Reviewer 3 Report
None
Author Response
Thank for your reviewer.